# Impact of Five Weeks of Strengthening Under Blood Flow Restriction (BFR) or Supplemental Oxygen Breathing (Normobaric Hyperoxia) on the Medial Gastrocnemius

**DOI:** 10.3390/jfmk9040258

**Published:** 2024-12-05

**Authors:** Grégory Vervloet, Lou Fregosi, Arthur Gauthier, Pierre Grenot, Costantino Balestra

**Affiliations:** 1Environmental, Occupational, Aging (Integrative) Physiology Laboratory, Haute Ecole Bruxelles-Brabant (HE2B), 1160 Brussels, Belgium; lou.fregosi06@gmail.com (L.F.); arthurgauthier888@gmail.com (A.G.); pgrenot@he2b.be (P.G.); cbalestra@he2b.be (C.B.); 2Anatomical Research and Clinical Studies, Vrije Universiteit Brussels (VUB), 1090 Brussels, Belgium; 3Motor Sciences Department, Physical Activity Teaching Unit, Université Libre de Bruxelles (ULB), 1050 Brussels, Belgium; 4DAN Europe Research Division (Roseto-Brussels), 1160 Brussels, Belgium

**Keywords:** training, hyperoxia, blood flow restriction (BFR), ischemia, force, medial gastrocnemius, ultrasound, 3/7 method, isometric, dynamic, human, oxygen

## Abstract

**Background and Objectives:** This study investigates the effects of a five-week training program on the medial gastrocnemius muscle, comparing two approaches: blood flow restriction (BFR) training and normobaric hyperoxia (oxygen supplementation). It evaluates three strengthening modalities (dynamic, isometric, and the 3/7 method) analyzing their impact on maximal voluntary contraction (MVC), muscle architecture, and perceived exertion. **Methods:** A total of 36 young healthy participants (21 females, 15 males) were randomized into six subgroups (n = 6 each) based on the type of contraction and oxygen condition. Training sessions (three per week) were conducted for five weeks at 30% of MVC. Measurements of MVC, muscle circumference, pennation angle, fascicle length, and perceived exertion were taken at baseline (T0), mid-protocol (T1), and post-protocol (T2). **Results:** All groups demonstrated significant increases in MVC after five weeks, with no notable differences between BFR and oxygen conditions. Structural changes were observed in specific subgroups: the BFR-isometric group showed increased calf circumference (*p* < 0.05), and the 3/7 groups exhibited significant fascicle length gains (*p* < 0.05). Perceived exertion was consistently higher in BFR groups compared to oxygen supplementation, particularly in dynamic exercises. **Conclusions:** Both BFR and oxygen supplementation are effective in enhancing strength with light loads, though they elicit different structural and perceptual responses. Oxygen supplementation may be more comfortable and less strenuous, offering a viable alternative for populations unable to tolerate BFR. Future research should focus on optimizing training parameters and exploring applications tailored to specific athletic or clinical contexts.

## 1. Introduction

Muscle strengthening plays a vital role in various physical activities and sports, with specific objectives varying depending on the individual’s goals, whether athletic performance or rehabilitation [1,2,3]. Individuals respond differently to different types of strength training, highlighting the need for personalized approaches [4]. Strength-building programs typically focus on low repetition counts with heavy loads, usually ranging from 60% to 90% of maximal voluntary contraction (MVC), while endurance training involves higher repetition counts with lighter loads, around 20% of MVC [5]. The intensity and volume of training, including rest intervals, significantly impact both acute responses and long-term adaptations in strength programs [6]. The 3/7 method is a strength-building approach that includes five sets of increasing repetitions (3, 4, 5, 6 and 7), performed with a 65–70% load relative to one-repetition maximum (RM) and a short 15 s rest interval. This method promotes metabolic buildup, tissue oxygenation deficits, and neuromotor adaptations, contributing to muscle hypertrophy and strength gains [7]. A modified version by Stagier et al. (2018) includes two cycles of five sets with a 90 s rest between cycles, designed to maintain repetition consistency [8]. Dynamic exercises, involving full range-of-motion movements with both concentric and eccentric contractions, are effective in enhancing strength and endurance while reducing muscle fatigue [9,10]. These exercises engage multiple muscle fibers and are beneficial for overall muscular development. In contrast, isometric exercises involve static muscle contractions without movement. When performed with high repetitions and appropriate loading, isometric exercises improve strength across specific ranges of motion and reduce muscle fatigue. Additionally, isometric training has been shown to benefit dynamic performance, such as in running, jumping, and cycling, with optimal contraction durations of 6 s per repetition [11].

Blood flow restriction (BFR) training, developed by Yoshiaki SATO in Japan during the 1970s and 1980s, involves the use of external compression (e.g., tourniquets or cuffs) to limit blood flow to muscles. This creates local ischemia and hypoxia, promoting muscle hypertrophy even with light loads, typically around 30% of MVC. BFR training has proven beneficial in clinical populations, such as those recovering from ACL surgery or suffering from femoro patellar syndrome, by reducing pain and improving function [12,13,14,15,16,17]. While less effective than heavy-load training in absolute terms, BFR is more tolerable, making it a promising option for early-stage rehabilitation [18]. Exercise, particularly under normoxic conditions, induces muscle damage and inflammation through myokine production (e.g., IL-6) and the activation of immune cells like macrophages and neutrophils, which support muscle regeneration and hypertrophy [19,20,21,22,23,24,25,26]. BFR training, by inducing hypoxia, also activates immune responses via HIF-1α transcription factors, which play a key role in muscle growth [27,28,29,30]. Recent studies have also suggested that similar immune responses can be triggered under hyperoxic conditions, especially following fluctuations between hyperoxia and normoxia, which may stimulate gene expression changes associated with the normobaric oxygen paradox (NOP) [21,22]. These processes indicate that both BFR and hyperoxic training may provoke comparable inflammatory and immune responses.

Training in hypoxic conditions has been well-documented for its effects on immune and inflammatory responses. Similar effects have been suggested in hyperoxic conditions, where oxygen fluctuations can trigger cellular responses. For example, hyperoxic exposure followed by a return to normoxia may induce the NOP, activating genes regulated by HIF-1α, which are typically responsive to hypoxia [22]. Inflammation markers such as IL-6 and IL-10 can be detected even at oxygen levels as low as 30% FiO_2_ after hyperoxic exercise [31]. Recent research shows also the effect of NOP on redox-mediated PGC1α-NRF1-2 signaling, assessed by the upregulation of transcription factor A, mitochondrial (TFAM) [20].

Despite the promising benefits of both BFR and hyperoxic training, the research on hyperoxic training is still limited, particularly regarding its underlying mechanisms and its comparison with BFR. This study aims to address this gap by comparing a 5-week hyperoxic training program (three sessions per week) targeting the medial gastrocnemius muscle in a healthy young population to BFR training. We hypothesize that both training modalities will lead to similar improvements in strength and muscle volume. Additionally, we propose that hyperoxic training may offer greater comfort and accessibility, with fewer contraindications compared to BFR.

## 2. Materials and Methods

This study investigated the effects of different strengthening methods and oxygenation conditions on the medial gastrocnemius muscle. Participants were recruited from a pool of healthy physiotherapy students between 1 September 2022, and 31 August 2023. All participants volunteered for the study after learning about it through social media and an information session at the physiotherapy school. They were fully briefed on the study’s objectives, potential risks, and benefits, and each provided written informed consent. The study follows a prospective design and divides participants into three main groups (isometric, dynamic, and 3/7 method), each further divided into two subgroups (oxygen and BFR) of six randomized subjects. We used reliable, straightforward, and non-invasive measurements to assess the strengthening program’s efficacy by tracking structural and physiological adaptations through various parameters.

### 2.1. Measurement of the Maximal Voluntary Isometric Contraction of the Dominant Calf Using a Digital Dynamometer

Subjects first completed a 5 min warm-up with a jump rope. Each participant then lay prone on a table designed to support contraction of the dominant leg’s calf muscle (Figure 1). Using a digital dynamometer (Mini Crane Scale MNCS-M; Matera, Italy) positioned beneath the table, the MVC was assessed by having participants perform a maximum 3 s contraction, repeated three times. This MVC measurement allowed us to calculate the training load, set at 30% of the MVC for all protocols [32]. Following recommendations from the literature, we maintained the intensity of the oxygen-supplemented strengthening protocol at 30% MVC [33,34]. After the eighth session, we re-evaluated MVC to adjust the load and prevent plateauing [35,36], acknowledging that untrained individuals may struggle to reach full MVC without progressive training [37].

### 2.2. Measurement of the Calf Circumference

The examiner measured the circumference at the largest curve of the dominant leg’s calf. Participants stood with the hip and knee of the measured side flexed at 90°, with the foot resting on a chair. Measurements were taken in centimeters (cm).

### 2.3. Measurement of the Pennation Angle and the Fascicule Length with Ultrasound

Two parameters were measured from each ultrasound scan using a Mindray DP-2200 ultrasound scanner (Shenzhen, China): medial gastrocnemius fascicular length, and pennation angle. Those two parameters reflect structural adaptations in muscle.

The greater the increase in pennation angle, the more it reflects hypertrophy of the studied muscle. Conversely, a reduced pennation angle ensures efficient force transmission, which is particularly advantageous in sprinting and jumping sports. Fascicle length also demonstrates an adaptation to exercise, serving as an indicator of effective muscular work [38].

Contact gel was applied to ensure high-quality imaging. The ultrasound probe was positioned one-third of the distance from the knee bend to the medial malleolus with minimal pressure.

The muscle fascicle was defined as a clearly visible fiber bundle lying between the superficial and deep aponeuroses. The pennation angle was determined as the angle between the fascicle and its insertion on the deep aponeurosis.

With the participant lying on an examination table and the lower leg supported by a 20° foam wedge (knee flexed to 20–30°), we reduced ankle dorsiflexion, allowing the gastrocnemius to relax. This positioning was adapted from Kwah et al. to optimize the validity of ultrasound-based pennation angle measurements [39].

The fascicle length (Lf) was measured along the marked fibers’ bundle, from the superficial to the deep aponeurosis. When the end of the fascicle extended off the acquired ultrasound image, fascicle length (Lf) was estimated using trigonometry (total Lf = lf 1 measured + lf 2 estimated = lf 1 + (h/sin μ)) by assuming a linear continuation of the fascicles (Figure 2) [18,40].

### 2.4. Measurement of Perception Effort Using the Borg Scale

We used the modified Borg scale to assess the subjective intensity of the training weekly. This scale, which ranges from 6 to 20, allowed participants to rate their exertion level, providing feedback to researchers to make adjustments if necessary [41].

### 2.5. Participants and Experimental Protocol

A total of 36 young healthy volunteers (21 females, 15 males) meeting the inclusion criteria were randomly assigned to experimental groups.

Participants were eligible for inclusion if they met the following criteria:-They were of legal age and no older than 35 years;-They provided a signed and dated informed consent form;-They were in good health, with no contraindications for engaging in sports activities.

The exclusion criteria were as follows:-Presence of arterial calcifications;-Severe hypertension;-Renal pathologies;-History of deep vein thrombosis;-Use of anticoagulant drugs;-Cardiovascular pathologies;-Epilepsy;-History of pneumothorax;-Diabetes;-Recent musculoskeletal injuries, including lower limb sprain (<3 weeks) or fracture (<3 months);-Recent orthopedic surgery (<6 weeks).

The 5-week program involved three sessions per week, focusing on the heel rise test (HRT) as the strengthening exercise, as described by Monteiro et al. (2017) and Pires et al. (2020) [42,43]. The exercise was performed within a range of motion (ROM) from −25° to +25° dorsiflexion. Following Kassiano et al. (2023), we chose a full ROM procedure to optimize gastrocnemius adaptation compared to the limited 0 to 25° ROM [44,45]. Each eccentric phase was performed over 2 s to increase mechanical load.

Participants were randomly divided into three different groups:-The dynamic group; 4 × (30–15–15–15 reps) r = 30 s R = 90 s 1-0-2 [34].-The isometric group; (15–7–7–7) × 6 s isom r = 30 s R = 90 s [46].-The 3/7 method group; 2 × (3–4–5–6–7 reps) r = 15 s R = 90 s 1-0-2 [7].

In our study, ‘1-0-2’ describes the tempo imposed on participants during their strength training sessions and refers to a movement performed with a 1 s concentric phase (1), no pause at the end of the movement (0), and concluding with a 2 s eccentric phase (2).

Cuff placement and LOP for the BFR group followed recent protocols to customize pressure [16,33,47]. The limb occlusion pressure (LOP) was measured at the beginning of the experiment after the participant had been lying down for 1 min. The cuff was positioned as proximally as possible, at the root of the lower limb. An audible Doppler device (Figure 3) was used to detect the moment the LOP was reached, indicated by the disappearance of pulsations in the posterior tibial artery.

Every subject during BFR sessions was equipped with the compression device (H-Cuff) set at 60% of their own AOP for the three first sessions then 70% for the two following sessions and 80% for the two last sessions before the adaptation of the workload Figure 4). This was followed by a new progressive increase in the percentage of AOP, starting at 60% for the first 3 sessions, 70% for the next 2 sessions, and reaching 80% of AOP for the final 3 sessions of the cycle, with the loads adjusted based on the updated MVC measurement (Figure 5).

In all oxygen supplementation subgroups, participants follow the protocol using high levels of oxygen at 95% O_2_. An O_2_ concentrator and non-rebreather type oxygen mask has been used to this end during exercise. The experimental protocol is outlined in the timeline in Figure 5. All variables have been measured at T0, T1 and T2.

All experiments took place at He2B—ISEK, Avenue Charles Schaller 91, Auderghem, Belgium, from February to April 2023. The study protocol was approved by the Academic Ethical Committee of Brussels (B200-2021-200) and adhered to the Declaration of Helsinki.

### 2.6. Statistical Analysis

Conventional statistical methods were used via GraphPad Prism 9 for calculating the means, standard deviations (SD), and standard error (SE). The medians and quarters were also calculated. The Kolmogorov–Smirnov and the D’Agostino–Pearson omnibus tests were used to assess normality.

Changes in MVC, perimeter, evolution of the pennation angle, and fascicle length were analyzed.

The specific time points before and after specific training were analyzed by means of a Student’s *t*-test, or Mann–Whitney test when appropriate.

Cohen’s D with 95% CI was used to calculate the size effect.

Power calculation was performed a priori (effect size = 1.6, alpha error = 0.05, power = 0.80) using G*Power calculator 3.1 software (Heinrich Heine University, Düsseldorf, Germany); the requisite number of participants for this study was equal to 6 based on previous data [40].

## 3. Results

### 3.1. Physiological Parameters

Age, BMI, and hour per week of sport practice collected from all subjects are reported in Table 1. Data did not differ significantly.

### 3.2. Maximal Voluntary Isometric Contraction

#### 3.2.1. Intra-Group Comparison

The evolution of MVC is significant for every intra-group analysis after 5 weeks except for the isometric method in T1. The oxygen dynamic method appears to show the best results with a highly significant *p*-value at the end of the protocol (Table 2).

#### 3.2.2. Inter-Group Comparison

In the dynamic group, a significant difference was observed between the oxygen and BFR group (*p* = 0.0451), with the BFR group showing better results at the end of the fifth week of the protocol. For the 3/7 and isometric groups, neither the hypoxic nor hyperoxic conditions resulted in better outcomes in the evolution of gastrocnemius’ MVC (Table 3).

### 3.3. Calf Circumference

#### 3.3.1. Intra-Group Comparison

As depicted in Table 4, the only group in which calf circumference showed significant progress after 3 weeks (*p* = 0.0272 + 3%) and 5 weeks (*p* = 0.0464 + 4%) of training was the isometric BFR group. None of the other groups showed significant differences after following the protocol.

#### 3.3.2. Inter-Group Comparison

No group shows any significant results after the protocol of the study (Table 5).

### 3.4. Pennation Angle

#### 3.4.1. Intra-Group Comparison

No significant results have been observed in the evolution of the pennation angle except in the isometric oxygen group, where there is an 11% increase in the angle (*p* = 0.0436 *), as shown in Table 6.

#### 3.4.2. Inter-Group Comparison

No significant differences have been observed in any of the groups or subgroups regarding changes in the pennation angle over time (Table 7).

### 3.5. Fascicule Length

#### 3.5.1. Intra-Group Comparison

As indicated in Figure 6 and Table 8 after five weeks of training, only both subgroups within the 3/7 group showed significant results in fascicle length.

#### 3.5.2. Inter-Group Comparison

There were no notable findings seen following the five weeks of protocol training comparing the two subgroups across all contraction regimes (Table 9).

### 3.6. Borg Scale

Table 10 displays the mean values reported by our subjects along the experiment. The *p* values show significant and very significant differences between the oxygenic conditions. The BFR groups show the most exertional scores in the both dynamic protocols (dynamic and 3/7).

Table 11 displays changes in perceived effort during the protocol, as measured by the Borg score. It is evident that the BFR subgroup experienced greater difficulty in performing the exercise during the first session compared to the oxygen subgroup across all three strengthening groups.

A clear difference in perceived fatigue is shown between the two modalities (oxygen and BFR); however, it seems that the 3/7 protocol is the one that most consistently maintains fatigue levels from S1 to S15, whereas in the other two groups (dynamic and isometric), the difficulty of performing the exercise appears to decrease.

In contrast, the dynamic training group exhibited a high Borg score at the beginning of the study and maintained high values throughout. This is different from the other training groups (3/7 and isometric), where there was a decrease in the Borg score over the course of the training sessions. This trend is associated with the complaints and DOMS reported by the subjects during the protocol, which were more prevalent in the dynamic group.

### 3.7. Crossover Effect on the Maximal Voluntary Isometric Contraction of the Contralateral Leg

#### Intra-Group Comparison

As shown in Figure 7 and Table 12, it has been noted that significant to very significant changes have occurred in all subgroups in terms of crossover MVC, except for the BFR dynamic group throughout the duration of the protocol (Figure 7).

## 4. Discussion

Exercise at 30% of MVC is typically not classified as strength training under normoxic conditions, even with standard set and repetition schemes. Strength gains are generally observed beginning at 60–70% of MVC with conventional training protocols [5,48].

Given this threshold, it was anticipated that groups supplemented with oxygen would display similar effects at the 30% MVC workload. However, training at this intensity with blood flow restriction (BFR) creates a challenging workout due to the metabolic stress induced by restricted venous return and partial arterial flow [25,28].

Following the second MVC test, all groups demonstrated improved T1 results compared to T0. These improvements were statistically significant in the 3/7 contraction groups with oxygen and BFR training (*p* = 0.0387 * and *p* = 0.0181 *, respectively) as well as in the dynamic contraction groups with oxygen and BFR training (*p* = 0.0206 * and *p* = 0.0312 *, respectively). The only exception was the isometric contraction group, which did not show significant improvement, likely due to high baseline values at T0 [37].

Across groups, no significant differences in outcomes were found between different oxygen conditions at T1. This lack of difference may stem from neural adaptations improving motor unit recruitment and synchronization, optimizing movement and agonist–antagonist coordination [49]. These adaptations allow for efficient agonist muscle contraction by reducing antagonist activation through reciprocal inhibition, which can otherwise impede force production [50].

At the study’s conclusion (T2), all groups exhibited significant intra-group MVC improvements, with the exception of the dynamic group, where the BFR group showed more pronounced progress than the oxygen group. This difference may be due to the dynamic group’s higher repetition count and longer rest periods, which increased workload intensity, especially under the combined influence of BFR and extended occlusion time. While the BFR and oxygen subgroups showed no statistically significant differences across other contraction modes, both methods were effective in strength gains, particularly in dynamic exercises. Balestra et al. (2021) noted comparable metabolic stress in both BFR and normobaric oxygen protocols, suggesting that “cellular training” creates similar effects by inducing metabolic stress upon return to normoxia, paralleling BFR effects [31].

Program duration influences observed results and interpretation, as neural adaptations occur faster than structural ones. Knight et al. (2001) demonstrated that neural adaptations in normoxic conditions appear within the first four weeks of training, with structural adaptations emerging around the eighth week [51]. In hypoxic conditions (BFR), 12-week protocols using equivalent load routines in BFR and control groups showed cell swelling and strength gains [13], with MVC increases appearing as early as 6 weeks [52] or even 4 weeks in BFR groups compared to controls [53]. In a 5-week protocol, Balestra et al. (2021) observed similar improvements through varied inspired oxygen levels [31]. Laurent et al. (2016) observed a 29.8% MVC improvement with an 8-week 3/7 method protocol (without BFR) performed twice weekly, and shorter protocols (1–3 weeks) also indicated cell swelling and strength gains [54].

Although our study duration (5 weeks) was shorter than long-term protocols, MVC gains were observed within 2.5 weeks (seven sessions) [31,53].

Neither the 3/7 nor dynamic method groups experienced significant calf circumference changes, while the BFR isometric group did (T0–T1, *p* = 0.0272 *; T0–T2, *p* = 0.0464 *). However, no significant difference was observed between oxygen conditions by T2 (*p* = 0.0610). This intra-group increase aligns with findings from Lauber et al. (2021), who showed that BFR isometric training produces more intramuscular metabolites than dynamic training, likely due to increased metabolic stress from venous stasis induced by the BFR cuff [46].

For other groups, similar results were noted by Shiromaru (2019), where no significant changes occurred in calf circumference following a 6-week low-load BFR protocol, indicating no muscle edema or degradation [55].

None of the oxygen groups in any method showed significant differences, likely due to the light load used and the lack of additional metabolic stress in a short protocol (5 weeks), which may not have been enough to stimulate measurable structural adaptations in calf circumference.

There were no significant differences in calf circumference between the oxygen supplementation and BFR groups at T2 after training. In other words, neither method proved to be more effective than the other in producing hypertrophy as measured by means of a tape after 5 weeks.

Training did, however, increase fascicle count and thickness [56,57]. Pennation angle, indicative of hypertrophy, was affected only in the oxygen isometric group (*p* = 0.0436 *), potentially due to rapid cellular responses, as recently observed in TFAM levels after oxygen exposure [20].

Interestingly, the similarity between training and evaluation positions in the isometric group could have contributed to increased metabolic stress and structural adaptations [46,58].

Fascicle length notably increased in the 3/7 subgroups (BFR 3/7, *p* = 0.0161 *; O_2_ 3/7, *p* = 0.0483 *) without significant pennation angle changes. This is consistent with Geremia et al. (2019), who suggested that tempo and range of motion can enhance fascicle length without affecting pennation angle [59].

Stragier et al. demonstrated similar results under normoxia in the short term [7,60]. The addition of metabolic stress via occlusion or normobaric oxygen supplementation (NOP) may enhance outcomes, as suggested in prior research [31,61,62].

The lack of significant fascicle length increases in other groups may be due to protocol differences, as eccentric training appears more beneficial for fascicle length adaptation, a component not present in our isometric protocol [59]. Higher repetition in the dynamic group led to partial range-of-motion completion due to fatigue, despite monitoring, potentially contributing to outcome differences [63].

The external validity of BFR studies is also noteworthy; much of the research is based on populations with pathological conditions [14,15,29,34,64,65]. Consequently, these populations may have had a higher progression potential compared to our healthy, pre-screened participants.

Our graphical analysis shows a progressive adaptation to effort among participants. Stragier et al. attributed this to decreasing inflammatory responses to muscle damage during early training, irrespective of oxygen or BFR conditions [7].

BFR participants also underwent gradual cuff pressure increases, starting at 60% AOP for three sessions, then progressing to 70% and 80% across sessions. Strength gains were confirmed through significant ΔT1 increases across groups, with MVC levels rising after seven sessions and corresponding increases in perceived effort by the Borg scale in the eighth session.

Variability in weekly sport hours among participants influenced perceived effort, with isometric group participants (engaging in less weekly sport) perceiving training as more challenging. Since the participants were physiotherapy students, school schedule, fatigue, and motivation over the 5-week protocol may have also impacted outcomes. BFR training generally elicited higher effort perception, as unfamiliarity with occlusion contributed to discomfort, echoing Suga et al.’s 2021 findings on BFR’s impact on pain, motivation, and exercise enjoyment [66]. Athletes accustomed to challenging sensations may better tolerate BFR, as Kataoka et al. indicated [67].

When elite athletes perform low-load strength training with BFR, it allows them to replicate the intense sensations experienced during regular high-intensity training. This simulation can help sustain motivation and commitment during “deload” periods where training loads are reduced by approximately 25–50% to allow active recovery without reducing training frequency, which not only helps mitigate injury risk but also optimizes recovery and physical conditioning gains [68].

However, leading up to key events, athletes may find it challenging to experience the same intensity as in high-load training. For this reason, combining low-load strength training with BFR may be particularly beneficial for elite athletes. R.J. Wortman et al. (2021) found that BFR added to traditional training enhances performance by simulating a more intense training experience [69]. Based on the Borg scale, BFR programs were perceived as more demanding across all strengthening types, suggesting that BFR during low-load training can sustain athlete motivation and adherence to periodized programs. It indicates also that BFR training is perceived as more strenuous than oxygen supplementation across all three training modes. Consequently, hyperoxia may be a viable alternative for enhancing muscle strength in those unable to tolerate heavy loads or BFR-induced occlusion.

Strength training can result in structural hypertrophy, which may increase muscle mass and body weight [70,71]. However, our findings show system adaptations affecting motor command, without structural changes. These adaptations are beneficial in weight-sensitive sports (e.g., combat sports, weightlifting, gymnastics), where athletes can maintain or enhance strength (as seen in MVC evaluations) without increasing body mass. This is particularly advantageous during the recovery or downtime phases when weight gain may impact performance, such as in the period between a weigh-in and competition.

Our study confirmed improvements in the maximal voluntary contraction (MVC) of the dominant leg gastrocnemius across all training groups. Similar to Bowman et al. (2019), who found strength gains in the non-dominant leg after unilateral BFR training, our results suggest a comparable crossover effect, as MVC of the non-dominant leg improved across all groups after 5 weeks [72]. Prior studies have shown that unilateral increases iEMG and motor unit recruitment in the contralateral limb, a potential asset in rehabilitation where targeted strengthening of one limb could benefit the opposite limb through BFR or oxygen supplementation [73,74].

Nevertheless, these findings should be cautiously used because our study was conducted in a population of healthy, young individuals.

Exercise selection should consider specific goals (e.g., strength vs. hypertrophy) and individual preferences. Only the isometric group exhibited structural changes, with increased pennation angle and calf circumference, possibly due to the alignment of evaluation and training positions. Significant fascicle length increases were observed in the 3/7 groups, likely facilitated by optimal eccentric loading. Importantly, all groups demonstrated improved MVC over the 5-week protocol regardless of oxygen or BFR conditions, suggesting that the gains were predominantly driven by neural adaptations rather than structural hypertrophy, as discussed earlier.

## 5. Practical Applications and Strengths of the Study

The practical applications and strengths of the current study are as follows:-The blood flow restriction (BFR) cuff placement protocol was individualized, taking into account arterial occlusion pressure (AOP) and limb occlusion pressure (LOP) [34,65,75].-All participants used identical equipment, including high-concentration oxygen masks, weight materials, and cuffs (H-Cuff) [47,76].-A second MVC test was conducted after the seventh session to establish a new MVC and adjust the workload. Metabolic stress was also incrementally increased by progressively raising the BFR pressure throughout the training period [34,39,46].

To the best of our knowledge, this is the first study to randomly target and compare the effects of a specific strength training protocol across two different oxygen conditions. Despite employing low loads, even in hyperoxic conditions, our results demonstrate gains in MVC, providing an alternative approach for individuals who may not tolerate BFRT or who require a temporary reduction in training intensity for clinical or periodization purposes. Previous research, such as Barbalho et al. (2019), has examined the application of BFRT in patients with restricted mobility, such as those in comas. Given the routine use of oxygen therapy in hospitals and its fewer contraindications compared to BFRT [77], our findings may suggest potential applications in clinical settings, warranting further investigation.

In clinical practice, these findings may be relevant for patients undergoing rehabilitation, such as in cancer recovery programs, as exercise is increasingly integrated into medical treatments. Patient adherence and active engagement are essential for successful outcomes in such cases [78].

Furthermore, our protocol may also be of interest to athletes in weight-class sports, where increasing MVC without concurrent weight gain is advantageous.

## 6. Limitations

This study has several limitations. First, the absence of a control group limits the ability to directly compare the observed outcomes. Additionally, strengthening sessions were not supervised on weekends, which may have influenced consistency in training. A comparison of the mechanical and architectural parameter behavior between males and females was not conducted but should be considered in future studies, as differences in perceptions and tolerance to strengthening exercises might exist.

Future studies would be valuable to explore how a patient’s predisposition toward anaerobic or aerobic activities might impact the outcomes of such a protocol.

## 7. Conclusions

The present study aimed to compare the effectiveness of blood flow restriction (BFR) and oxygen supplementation in inducing changes in muscle architecture and strength gains in the gastrocnemius muscle among young, healthy individuals using three different strengthening methods. The results indicate no significant difference between BFR and oxygen supplementation in enhancing strength over the 5-week training period, irrespective of the specific strengthening program employed. Structural adaptations were observed in the 3/7 and isometric groups following the training period.

Both BFR and oxygen supplementation may be beneficial during early rehabilitation stages with light loads, promoting physiological and neurophysiological adaptations at the muscle level. They may also be useful during the “deload” phase in athletes, where reduced training intensity is required. However, the observed effects may not be as pronounced in young, active individuals.

Further research is needed to identify the optimal training load and exposure parameters that maximize gains with oxygen supplementation. Additionally, exploring the effects of BFR and oxygen supplementation in relation to the type of sport practiced and individual preferences would provide valuable insights, especially considering the specific characteristics of the athletes involved in the study.

## Figures and Tables

**Figure 1 jfmk-09-00258-f001:**
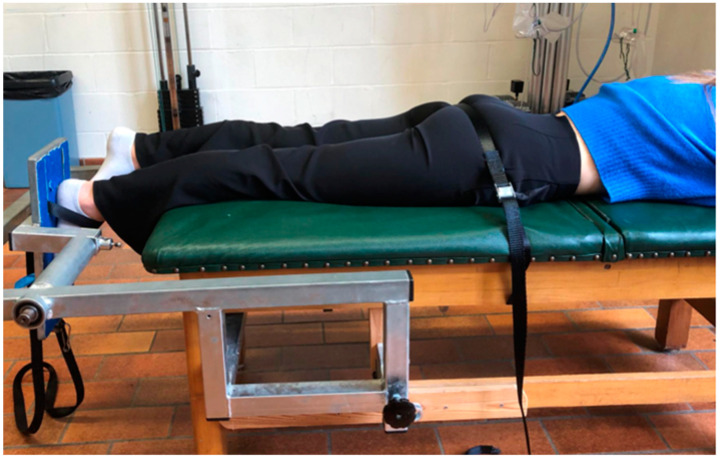
System set up for the MVC test.

**Figure 2 jfmk-09-00258-f002:**
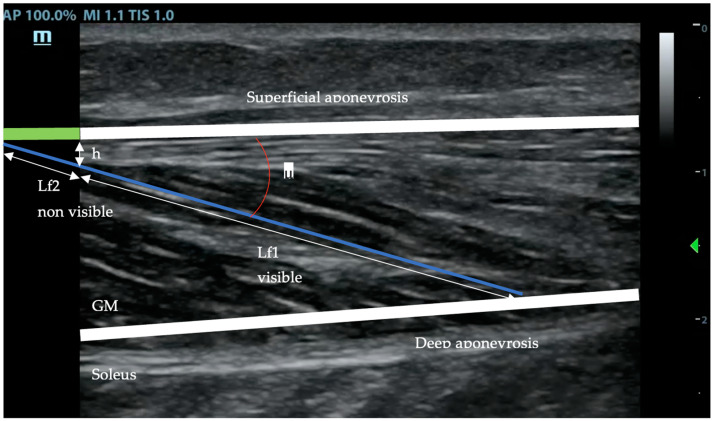
Ultrasound image showing the measures taken to fill in the formula for the fascicule total length: total (Lf) = lf1 + lf2 = lf1 + (h/sin μ). This approach is necessary since direct full measurement of fascicule length is out of the probe vision.

**Figure 3 jfmk-09-00258-f003:**
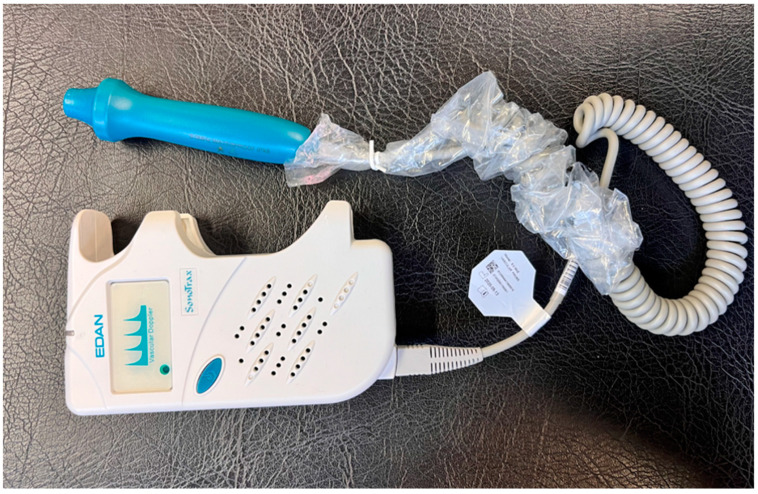
Audible Doppler device used to determine LOP.

**Figure 4 jfmk-09-00258-f004:**
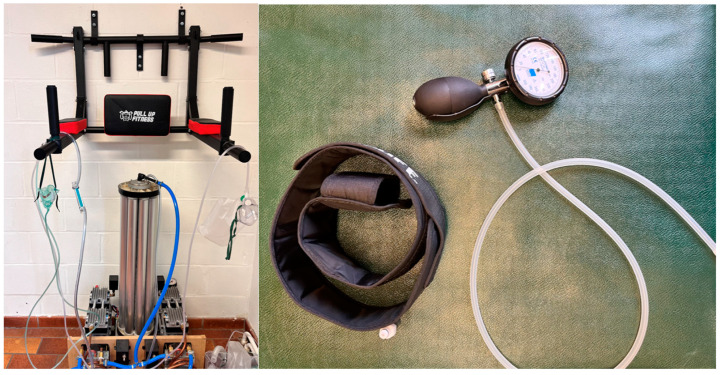
On the left, O_2_ concentrator and non-rebreather type oxygen mask; on the right, H-Cuff BFR model and pressure system.

**Figure 5 jfmk-09-00258-f005:**
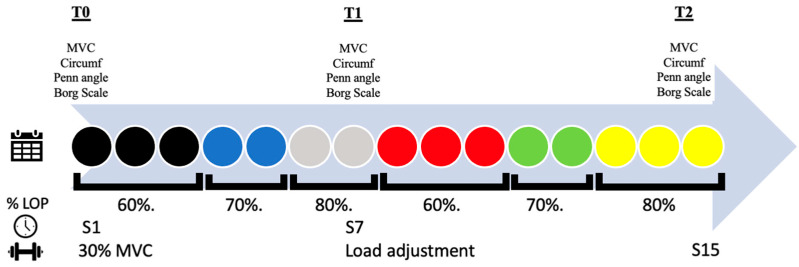
Experimental protocol design. As indicated, all subjects were tested at the first session (S1) (T0), the seventh session (S7) (T1), and at the end of the protocol (fifteenth session—S15) (T2). %AOP, arterial obliteration pressure percentage; MVC, maximal voluntary contraction; Circumf, circumference; Penn angle, pennation angle.

**Figure 6 jfmk-09-00258-f006:**
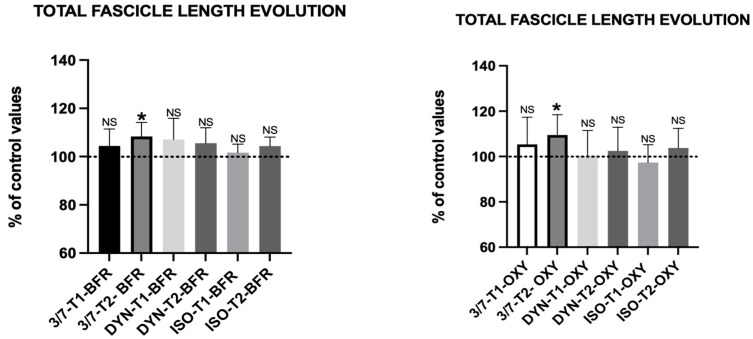
Effect of the protocol on the fascicule length of the gastrocnemius medius in the different contraction groups after 3 weeks (T1) and at the end of the protocol (T2) in both oxygenation conditions. Results are expressed as mean ± SD. Statistically significant differences symbols: NS non-significant, * *p* < 0.05.

**Figure 7 jfmk-09-00258-f007:**
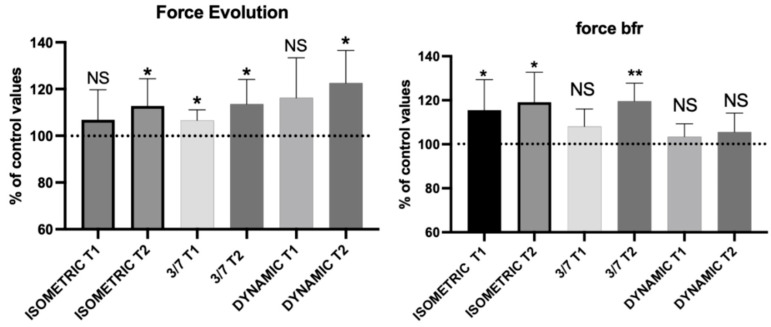
Effect of the protocol on contralateral leg MVC in the different contraction groups after 3 weeks (T1) and at the end of the protocol (T2) in both oxygenation conditions. Results are expressed as mean ± SD. Statistically significant differences symbols: NS non-significant, * *p* < 0.05, ** *p* < 0.01.

**Table 1 jfmk-09-00258-t001:** Physiological parameters collected from all subjects. BMI: body mass index. Data are expressed as mean values (±SD).

**3/7**			
	**BFR**	**Oxygen**	***p*-Value**
**Age (year)**	21 ± (1.9)	21 ± (1.6)	*p* = 0.7056
**BMI**	23 ± (2.3)	24 ± (2.8)	*p* = 0.5917
**Sport (h/wk)**	5.0 ± (2.0)	5.2 ± (3.1)	*p* = 0.8960
**Gender**	4W/2M	3W/3M	*p* > 0.9999
**Dynamic**			
	**BFR**	**Oxygen**	***p*-Value**
**Age (year)**	22.5 ± (2.258)	20.33 ± (1.033)	*p* = 0.0583
**BMI**	24.87 ± (2.229)	23.02 ± (2.708)	*p* = 0.2254
**Sport (h/wk)**	4.833 ± (2.317)	4.667 ± (2.875)	*p* = 0.9141
**Gender**	4W/2M	3W/3M	*p* > 0.9999
**Isometric**			
	**BFR**	**Oxygen**	***p*-Value**
**Age (year)**	21.33 ± (1.211)	20.00 ± (1.095)	*p* = 0.0734
**BMI**	21.81 ± (2.871)	20.76 ± (0.8992)	*p* = 0.4113
**Sport (h/wk)**	3.167 ± (1.472)	2.667 ± (1.366)	*p* = 0.5556
**Gender**	3W/3M	4W/2M	*p* > 0.9999

**Table 2 jfmk-09-00258-t002:** Intra-group evolution of the maximal voluntary contraction and delta for each interval collected from all subjects. Data are expressed as mean values (±SD), and as median and quarters. Cohen’s D was used to calculate the size effect with 95% CI. (**A**) Evolution of the maximal voluntary contraction for the interval T0–T1; (**B**) evolution of the maximal voluntary contraction for the interval T0–T2. (* *p* < 0.05; ** *p* < 0.01).

*Maximal Voluntary Contraction* ΔT0 T1 (**A**)					
**Condition**	**Group**	*Mean ± (SD)*	*Median (Q1; Q3)*	*p*-Value	*Cohen’s D*
Oxygen	3/7	108 ± (7.17)	*106.20 (103.39*; *114.30)*	*p* = 0.0387 *	0.116
BFR	3/7	109 ± (6.4)	*107.53 (105.16*; *109.72)*	*p* = 0.0181 *	0.0266
Oxygen	Dynamic	105 ± (3.88)	*105.25 (103.07*; *107.12)*	*p* = 0.0206 *	0.987
BFR	Dynamic	107 ± (3.1)	*106.01 (105.29*; *109.06)*	*p* = 0.0312 *	0.590
Oxygen	Isometric	113 ± (14.5)	*109.55 (107.23*; *117.05)*	*p* = 0.0733	0.288
BFR	Isometric	111 ± (12)	*110.85 (103.90*; *118.85)*	*p* = 0.0772	0.181
*Maximal Voluntary Contraction* ΔT0 T2 (**B**)					
**Condition**	**Group**	*Mean ± (SD)*	*Median (Q1; Q3)*	*p*-Value	*Cohen’s D*
Oxygen	3/7	110 ± (6.58)	*108.69 (107.43*; *110.04)*	*p* = 0.0129 *	*0.786*
BFR	3/7	116 ± (11)	*112.39 (107.26*; *124.54)*	*p* = 0.0184 *	*0.0755*
Oxygen	Dynamic	109 ± (3.81)	*109.26 (106.69*; *111.60)*	*p* = 0.0026 **	*1.619*
BFR	Dynamic	115 ± (5.1)	*112.66 (112.13*; *113.30)*	*p* = 0.0312 *	*0.0333*
Oxygen	Isometric	118 ± (14.4)	113.60 (109.78; 120.35)	*p* = 0.0298 *	0.197
BFR	Isometric	123 ± (15)	126.15 (123.88; 129.33)	*p* = 0.0139 *	0.522

**Table 3 jfmk-09-00258-t003:** Inter-group strength evolution and delta for each interval collected from all subjects. MVC: maximal voluntary contraction. Data are expressed as mean values (±SD) and as median and quarters. Cohen’s D was used to calculate the size effect with 95% CI. (* *p* < 0.05).

	BFR		Oxygen			
**Maximal Voluntary Contraction 3/7**						
**Interval**	**Mean ± (SD)**	**Median (Q1; Q3)**	**Mean ± (SD)**	**Median (Q1; Q3)**	***p*-value**	**Cohen’s D**
**ΔT0–T1**	109 ± (6.4)	*107.53 (105.16*; *109.72)*	108 ± (7.17)	*106.20 (103.39*; *114.30)*	*p* = 0.8230	*−0.29*
**ΔT0–T2**	116 ± (11)	*112.39 (107.26*; *124.54)*	110 ± (6.58)	*108.69 (107.43*; *110.04)*	*p* = 0.3019	0.66
**Maximal Voluntary Contraction Dynamic**						
**Interval**	**Mean ± (SD)**	**Median (Q1; Q3)**	**Mean ± (SD)**	**Median (Q1; Q3)**	***p*-value**	**Cohen’s D**
**ΔT0–T1**	107 ± (3.1)	*106.01 (105.29*; *109.06)*	105 ± (3.88)	*105.25 (103.07*; *107.12)*	*p* = 0.3220	*0.57*
**ΔT0–T2**	115 ± (5.1)	*112.66 (112.13*; *113.30)*	109 ± (3.81)	*109.26 (106.69*; *111.60)*	*p* = 0.0451 *	*1.33*
**Maximal Voluntary Contraction Isometric**						
**Interval**	**Mean ± (SD)**	**Median (Q1; Q3)**	**Mean ± (SD)**	**Median (Q1; Q3)**	***p*-value**	**Cohen’s D**
**ΔT0–T1**	111 ± (12)	*110.85 (103.90*; *118.85)*	113 ± (14.5)	*109.55 (107.23*; *117.05)*	*p* = 0.7743	*−0.15*
**ΔT0–T2**	123 ± (15)	*126.15 (123.88*; *129.33)*	118 ± (14.4)	*113.60 (109.78*; *120.35)*	*p* = 0.5443	*0.34*

**Table 4 jfmk-09-00258-t004:** Intra-group evolution of the circumference of the calf and delta for each interval collected from all subjects. Data are expressed as mean values (±SD) and as median and quarters. Cohen’s D was used to calculate the size effect with 95% CI. (**A**) Evolution of the circumference of the calf for the interval T0–T1; (**B**) evolution of the circumference of the calf for the interval T0–T2. (* *p* < 0.05).

*Circumference* ΔT0 T1 (**A**)					
**Condition**	**Group**	*Mean* ± *(SD)*	*Median (Q1; Q3)*	*p*-value	*Cohen’s D*
Oxygen	3/7	100 ± (1.2)	*99.86 (98.99*; *100.99)*	*p* = 0.9355	1.108
BFR	3/7	101 ± (1.9)	*101.44 (100.35*; *102.17)*	*p* = 0.3045	0.174
Oxygen	Dynamic	100 ± (3.2)	*100.11 (96.94*; *101.58)*	*p* = 0.8630	0.416
BFR	Dynamic	101 ± (3.3)	*100.02 (98.70*; *102.59)*	*p* = 0.6875	0.100
Oxygen	Isometric	103 ± (4.9)	*102.95 (100.35*; *106.98)*	*p* = 0.1919	0.341
BFR	Isometric	103 ± (2.4)	*102.35 (101.50*; *103.05)*	*p* = 0.0272 *	0.696
*Circumference* ΔT0 T2 (**B**)					
**Condition**	**Group**	*Mean* ± *(SD)*	*Median (Q1; Q3)*	*p*-value	*Cohen’s D*
Oxygen	3/7	100 ± (1.3)	*100.00 (99.80*; *100.00)*	*p* = 0.5922	*0.769*
BFR	3/7	100 ± (2.6)	*99.96 (97.57*; *102.30)*	*p* = 0.9541	*0.385*
Oxygen	Dynamic	101 ± (3.5)	*101.22 (98.17*; *102.82)*	*p* = 0.6050	*0*
BFR	Dynamic	101 ± (2.4)	*101.30 (101.26*; *102.25)*	*p* = 0.4375	*0*
Oxygen	Isometric	100 ± (2.9)	*98.60 (98.60*; *101.83)*	*p* = 0.9574	*0.345*
BFR	Isometric	104 ± (4,1)	*104.15 (102.03*; *107.70)*	*p* = 0.5922	*0.769*

**Table 5 jfmk-09-00258-t005:** Inter-group circumference evolution and delta for each interval collected from all subjects. Data are expressed as mean values (±SD) and as median and quarters. Cohen’s D was used to calculate the size effect with 95% CI.

	BFR		Oxygen			
**Circumference 3/7**						
**Interval**	**Mean ± (SD)**	**Median (Q1; Q3)**	**Mean ± (SD)**	**Median (Q1; Q3)**	***p*-value**	**Cohen’s D**
**ΔT0–T1**	101 ± (1.9)	*101.44 (100.35*; *102.17)*	100 ± (1.2)	*99.86 (98.99*; *100.99)*	*p* = 0.3372	*0.63*
**ΔT0–T2**	100 ± (2.6)	*99.96 (97.57*; *102.30)*	100 ± (1.3)	*100.00 (99.80*; *100.00)*	*p* = 0.8460	*0*
**Circumference Dynamic**						
**Interval**	**Mean ± (SD)**	**Median (Q1; Q3)**	**Mean ± (SD)**	**Median (Q1; Q3)**	***p*-value**	**Cohen’s D**
**ΔT0–T1**	101 ± (3.3)	*100.02 (98.70*; *102.59)*	100 ± (3.2)	*100.11 (96.94*; *101.58)*	*p* = 0.5453	*0.31*
**ΔT0–T2**	101 ± (2.4)	*101.30 (101.26*; *102.25)*	101 ± (3.5)	*101.22 (98.17*; *102.82)*	*p* = 0.9261	*0*
**Circumference Isometric**						
**Interval**	**Mean ± (SD)**	**Median (Q1; Q3)**	**Mean ± (SD)**	**Median (Q1; Q3)**	***p*-value**	**Cohen’s D**
**ΔT0–T1**	103 ± (2.4)	*102.35 (101.50*; *103.05)*	103 ± (4.9)	*102.95 (100.35*; *106.98)*	*p* = 0.9420	*0*
**ΔT0–T2**	104 ± (4.1)	*104.15 (102.03*; *107.70)*	100 ± (2.9)	*98.60 (98.60*; *101.83)*	*p* = 0.0610	*1.13*

**Table 6 jfmk-09-00258-t006:** Intra-group evolution of the pennation angle of the gastrocnemius medius and delta for each interval collected from all subjects. Data are expressed as mean values (± SD) and as median and quarters. Cohen’s D was used to calculate the size effect with 95% CI. (**A**) Evolution of the pennation angle of the gastrocnemius medius for the interval T0–T1; (**B**) evolution of the pennation angle of the gastrocnemius medius for the interval T0–T2. (* *p* < 0.05).

*Pennation Angle* ΔT0 T1 (**A**)					
**Condition**	**Group**	*Mean* ± *(SD)*	*Median (Q1; Q3)*	*p*-Value	*Cohen’s D*
Oxygen	3/7	101 ± (4.1)	*100.27 (99.76*; *104.23)*	*p* = 0.4588	0.0805
BFR	3/7	106 ± (7.0)	*105.92 (100.47*; *110.13)*	*p* = 0.1562	0.667
Oxygen	Dynamic	102 ± (16)	*99.04 (89.24*; *115.96)*	*p* = 0.7374	0.0419
BFR	Dynamic	98 ± (14)	*103.03 (88.06*; *105.96)*	*p* = 0.8617	0.238
Oxygen	Isometric	102 ± (11)	*101.90 (96.93*; *108.45)*	*p* = 0.6849	0.0609
BFR	Isometric	99 ± (8.1)	*100.75 (98.18*; *103.48)*	*p* = 0.7810	0.288
*Pennation Angle* ΔT0 T2 (**B**)					
**Condition**	**Group**	*Mean* ± *(SD)*	*Median (Q1; Q3)*	*p*-value	*Cohen’s D*
Oxygen	3/7	97 ± (2.8)	*96.71 (95.39*; *98.07)*	*p* = 0.0638	*2.261*
BFR	3/7	106 ± (5.7)	*107.52 (106.89*; *107.96)*	*p* = 0.0625	*0.468*
Oxygen	Dynamic	105 ± (13)	*99.04 (97.92*; *106.33)*	*p* = 0.4009	*0.128*
BFR	Dynamic	97 ± (16)	*98.18 (91.94*; *110.98)*	*p* = 0.7755	*0.396*
Oxygen	Isometric	111 ± (10)	*115.40 (106.60*; *118.20)*	*p* = 0.0436 *	*0.767*
BFR	Isometric	104 ± (7.9)	*104.05 (99.38*; *104.83)*	*p* = 0.2451	*0.0845*

**Table 7 jfmk-09-00258-t007:** Inter-group evolution of the pennation angle of the gastrocnemius medius and delta for each interval collected from all subjects. Data are expressed as mean values (±SD) and as median and quarters. Cohen’s D was used to calculate the size effect with 95% CI.

	BFR		Oxygen			
**Pennation angle 3/7**						
**Interval**	**Mean ± (SD)**	**Median (Q1; Q3)**	**Mean ± (SD)**	**Median (Q1; Q3)**	***p*-value**	**Cohen’s D**
**ΔT0–T1**	106 ± (7.0)	*105.92 (100.47*; *110.13)*	101 ± (4.1)	*100.27 (99.76*; *104.23)*	*p* = 0.3095	*0.87*
**ΔT0–T2**	106 ± (5.7)	*107.52 (106.89*; *107.96)*	97 ± (2.8)	*96.71 (95.39*; *98.07)*	*p* = 0.0649	*2*
**Pennation angle Dynamic**						
**Interval**	**Mean ± (SD)**	**Median (Q1; Q3)**	**Mean ± (SD)**	**Median (Q1; Q3)**	***p*-value**	**Cohen’s D**
**ΔT0–T1**	98 ± (14)	*103.03 (88.06*; *105.96)*	102 ± (16)	*99.04 (89.24*; *115.96)*	*p* = 0.7064	*−0.19*
**ΔT0–T2**	97 ± (16)	*98.18 (91.94*; *110.98)*	105 ± (13)	*99.04 (97.92*; *106.33)*	*p* = 0.4386	*−0.20*
**Pennation angle Isometric**						
**Interval**	**Mean ± (SD)**	**Median (Q1; Q3)**	**Mean ± (SD)**	**Median (Q1; Q3)**	***p*-value**	**Cohen’s D**
**ΔT0–T1**	99 ± (8.1)	*100.75 (98.18*; *103.48)*	102 ± (11)	*101.90 (96.93*; *108.45)*	*p* = 0.6142	*−1.32*
**ΔT0–T2**	104 ± (7.9)	*104.05 (99.38*; *104.83)*	111 ± (10)	*115.40 (106.60*; *118.20)*	*p* = 0.2231	*−0.78*

**Table 8 jfmk-09-00258-t008:** Intra-group evolution of the fascicule length of the gastrocnemius medius and delta for each interval collected from all subjects. Data are expressed as mean values (±SD). Cohen’s D was used to calculate the size effect with 95% CI. (**A**) Evolution of the fascicule length of the gastrocnemius medius for the interval T0–T1; (**B**) evolution of the fascicule length of the gastrocnemius medius for the interval T0–T2. (* *p* < 0.05).

*Fascicule Length* ΔT0 T1 (**A**)					
**Condition**	**Group**	*Mean* ± *(SD)*	*Median (Q1; Q3)*	*p*-value	*Cohen’s D*
Oxygen	3/7	105 ± (12)	*105.10 (99.09*; *109.90)*	*p* = 0.3260	0.208
BFR	3/7	104 ± (7.0)	*104.80 (99.04*; *108.26)*	*p* = 0.1846	0.214
Oxygen	Dynamic	100 ± (11)	*100.01 (92.60*; *102.77)*	*p* = 0.9341	0.227
BFR	Dynamic	107 ± (8.8)	*105.82 (102.27*; *114.38)*	*p* = 0.1036	0.511
Oxygen	Isometric	97 ± (7.9)	*95.10 (95.03*; *101.48)*	*p* = 0.4490	0.696
BFR	Isometric	102 ± (3.5)	*101.98 (99.42*; *103.51)*	*p* = 0.3125	0.143
*Fascicule Length* ΔT0 T2 (**B**)					
**Condition**	**Group**	*Mean* ± *(SD)*	*Med (Q1; Q3)*	*p*-value	*Cohen’s D*
Oxygen	3/7	109 ± (8.9)	*108.38* * (104.78*; *111.29)*	*p* = 0.0483 *	*0.393*
BFR	3/7	108 ± (5.8)	*108.94* * (104.45*; *112.81)*	*p* = 0.0161 *	*0.431*
Oxygen	Dynamic	102 ± (10)	*101.43* * (98.52*; *102.98)*	*p* = 0.5842	*0.35*
BFR	Dynamic	106 ± (6.4)	*106.08* * (104.58*; *106.48)*	*p* = 0.0862	*0.0781*
Oxygen	Isometric	104 ± (8.7)	*100.75* * (99.40*; *106.98)*	*p* = 0.3380	*0.172*
BFR	Isometric	104 ± (3.8)	*105.65* * (102.50*; *106.37)*	*p* = 0.0938	*0.395*

**Table 9 jfmk-09-00258-t009:** Inter-group fascicule length evolution. Data are expressed as mean values (±SD) and as median and quarters. Cohen’s D was used to calculate the size effect with 95% CI.

	BFR		Oxygen			
**Fascicule Length 3/7**						
**Interval**	**Mean ± (SD)**	**Median (Q1; Q3)**	**Mean ± (SD)**	**Median (Q1; Q3)**	***p*-value**	**Cohen’s D**
**ΔT0–T1**	104 ± (7.0)	*104.80** (99.04*; *108.26)*	105 ± (12)	*105.10** (99.09*; *109.90)*	*p* = 0.8717	*−0.10*
**ΔT0–T2**	108 ± (5.8)	*108.94** (104.45*; *112.81)*	109 ± (8.9)	*108.38** (104.78*; *111.29)*	*p* = 0.8074	*−0.13*
**Fascicule Length Dynamic**						
**Interval**	**Mean ± (SD)**	**Median (Q1; Q3)**	**Mean ± (SD)**	**Median (Q1; Q3)**	***p*-value**	**Cohen’s D**
**ΔT0–T1**	107 ± (8.8)	*105.82* * (102.27*; *114.38)*	100 ± (11)	*100.01** (92.60*; *102.77)*	*p* = 0.2724	*0.70*
**ΔT0–T2**	106 ± (6.4)	*106.08** (104.58*; *106.48)*	102 ± (10)	*101.43** (98.52*; *102.98)*	*p* = 0.5506	*0.48*
**Fascicule Length Isometric**						
**Interval**	**Mean ± (SD)**	**Median (Q1; Q3)**	**Mean ± (SD)**	**Median (Q1; Q3)**	***p*-value**	**Cohen’s D**
**ΔT0–T1**	102 ± (3.5)	*101.98** (99.42*; *103.51)*	97 ± (7.9)	*95.10** (95.03*; *101.48)*	*p* = 0.2453	*0.82*
**ΔT0–T2**	104 ± (3.8)	*105.65** (102.50*; *106.37)*	104 ± (8.7)	*100.75** (99.40*; *106.98)*	*p* = 0.8792	*0*

**Table 10 jfmk-09-00258-t010:** Inter-group Borg scale evolution as averages ± standard deviation. (* *p* < 0.05; ** *p* < 0.01).

Groups	Mean Values (±SD)	*p*-Value	Cohen’s D
Dynamic BFR	13.8 (±1.59)	*p* = 0.004 **	*1.15*
Dynamic O_2_	12.16 (±1.25)		
Isometric BFR	12.07 (±2.25)	*p* = 0.0809	*0.71*
Isometric O_2_	10.79 (±1.57)		
3/7 BFR	13.28 (±1.21)	*p* = 0.0234 *	*0.87*
3/7 O_2_	12.11 (±1.46)		

**Table 11 jfmk-09-00258-t011:** Intra-group Borg scale evolution between the first (S1) and last session (S15).

Groups	Mean Value S1	Mean Value S15	Evolution
Dynamic O_2_	14.8	11.3	−3.5
Dynamic BFR	16	11.1	−4.9
Isometric O_2_	13.3	9.33	−3.97
Isometric BFR	15	10	−5
3/7 O_2_	13.8	13.2	−0.6
3/7 BFR	13.5	13	−0.5

**Table 12 jfmk-09-00258-t012:** Intra-group contralateral MVC evolution as averages ± standard deviation. (* *p* < 0.05; ** *p* < 0.01).

**3/7**				
**Crossover MVC**	**ΔT0–T1**	***p*-value**	**ΔT0–T2**	***p*-value**
**Oxygen**	107 ± (4.49)	*p* = 0.0156	114 ± (10.5)	*p* = 0.0251 *
**BFR**	108 ± (8.0)	*p* = 0.0567	120 ± (8.2)	*p* = 0.002 **
**Dynamic**				
**Crossover MVC**	**ΔT0–T1**	***p*-value**	**ΔT0–T2**	***p*-value**
**Oxygen**	116 ± (17.1)	*p* = 0.0672	123 ± (13.9)	*p* = 0.0123 *
**BFR**	103 ± (5.8)	*p* = 0.2087	106 ± (8.6)	*p* = 0.1732
**Isometric**				
**Crossover MVC**	**ΔT0–T1**	***p*-value**	**ΔT0–T2**	***p*-value**
**Oxygen**	107 ± (12.9)	*p* = 0.2502	113 ± (11.7)	*p* = 0.0441 *
**BFR**	115 ± (14)	*p* = 0.0415	119 ± (14)	*p* = 0.0183 *

## Data Availability

All data are contained within the article.

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
