# Peer review of "Impact of Five Weeks of Strengthening Under Blood Flow Restriction (BFR) or Supplemental Oxygen Breathing (Normobaric Hyperoxia) on the Medial Gastrocnemius"

_jfmk, 2024, doi:10.3390/jfmk9040258_

Round 1
Reviewer 1 Report
Comments and Suggestions for Authors
Many thanks to the editor for giving me the opportunity to review this work. I hope my contribution will be useful.
The article is well written in describing the protocol, and the rationale behind it seems clear. It was interesting to read and evaluate it. There are a few notes I feel I must make and some questions to ask the authors:
Personal considerations:
- To clarify the two main groups better, I would give a more specific name to the oxygen supplementation group (just as it was done for the BFR group).
- Tables: It might be helpful to add asterisks to highlight a significant (*) or very significant (**) difference between the conditions.
Materials and Methods:
In this section, I would also include the set-ups of the different conditions with descriptions (if possible, an image/pictures), especially regarding blood flow restriction and oxygen supplementation.
It would be better to add a couple of lines for a more complete description of pennation angle and fascicle length parameters, considering the significance described later and why these parameters may be important.
Line 192, 194:
What do you mean with “1-0-2” in the description of the group?
Table 12:
As described in lines 314-317, it is evident that the BFR sub-group experienced more difficulty during the first session compared to the oxygen group. However, one clear point from this table is related to the training program.
Looking at the table, it seems that the 3-7 protocol is the one that most consistently maintains fatigue levels from S1 to S15, whereas in the other two groups (Dynamic and Isometric), the difficulty of performing the exercise appears to decrease. This could be a point to consider.
Line 513:
The adaptation of BFR to the individual subjects’ pressure characteristics should already be covered in the methods section, in a dedicated paragraph.
Limitation:
It could be considered a limitation that each group had only 6 subjects. Alternatively, it could be specified that 6 subjects can be a representative sample (if it can be demonstrated).
Reviewer 2 Report
Comments and Suggestions for Authors
Review jfmk-3343670-peer-review-v1
The aim of the paper Impact of five weeks of strengthening under blood flow restriction (BFR) or supplemental oxygen breathing (normobaric hyperoxia) on the medial gastrocnemius is to compare a 5-week, 3-session-per-week hyperoxic training program targeting the medial gastrocnemius muscle in a young, healthy population with BFR. The paper is interesting, but loses its value due to the lack of some information, the inconsistency of the markings, and the weak introduction. Below are the comments in order of occurrence rather than importance.
Abstract. Line 20 - there is no information about what are strengthening modalities (dynamic, isometric, and the 3/7 method), mainly the 3/7 method. Line 23 - please delete the age, and provide information that these were young people. Line 28 - please delete this information, because it is too general. It is better to not write anything.
Introduction. This chapter is poorly written. At the beginning, I felt that the sentences had no continuity of thought. You might want to consider dividing it into subsections that describe well-proposed methods, including their strengths and weaknesses. Then there will be a summary subsection that shows where the gaps in the literature are and places the need to present this work. Please consider the following points, you can of course combine them because I proposed a lot.
1. Muscle Strengthening Importance:
2. Strength and Endurance Programs:
3. The 3-7 Method:
4. Dynamic vs. Isometric Exercises:
5. Blood Flow Restriction (BFR) Training:
6. Scientific Mechanisms and Immune Responses:
7. Hypoxic vs. Hyperoxic Conditions:
8. Research Gap and Study Aim:
Material and Methods. This section is well written. Line 178 - please characterize women and men separately, do they train something? For Table 1 it is not clear what the exclusion criteria are and why they are in two columns. I think that instead of a table, it is better to write text or rework the table so that it is readable.
Line 191 - on what basis were the people divided into three groups, and by what method?
Figure 2 is not readable to me. What do the colors of the dots mean? What is %LOP, please describe all the markings. What is S1, S7, S15?
Section 2.6 – Please write exactly which parameters were compared using which tests. Why are parametric tests used with such a small group size? Was the a priori group size counted? Why are means and standard deviations and not medians and quarters reported in the results for non-parametric U-Mann, and Wilcoxon tests? This needs to be corrected. In my opinion, results should be presented using medians and quartiles.
Results. I don't understand the results in Tables 3, 5, 7, 9, and more specifically to what the p-values in the 3rd and last column refer. Please change the presentation of the results so that it is readable. I also remind you to switch to medians and quarters.
Discussion. This section is well written. On the other hand, it is necessary to add a study limitation.
Round 2
Reviewer 1 Report
Comments and Suggestions for Authors
Many thanks again to the editor for the opportunity to review this work.
As for the corrections made by me, the authors have complied with the requested changes, and it seems that the article can be accepted.
The inclusion of follow-up images and a more detailed description of the methods enhance the clarity and readability of the work, ensuring the reader has no uncertainties about the methodologies used
Reviewer 2 Report
Comments and Suggestions for Authors
The authors corrected the work and explained my doubts in detail. I have no additional comments. I think that the work can be accepted as it is. I do not hide the fact that it was difficult for me at some points to find my way through these corrections.